# Monkeypox Post-COVID-19: Knowledge, Worrying, and Vaccine Adoption in the Arabic General Population

**DOI:** 10.3390/vaccines11040759

**Published:** 2023-03-29

**Authors:** Sarya Swed, Haidara Bohsas, Hidar Alibrahim, Amine Rakab, Wael Hafez, Bisher Sawaf, Rais Mohammed Amir, Ahmed Sallam Motawei, Ahmed Aljabali, Sheikh Shoib, Ismail Atef Ismail Ahmed Ibrahim, Sondos Hussein Ahmad Almashaqbeh, Ebrahim Ahmed Qaid Shaddad, Maryam Alqaisi, Ahmed Abdelrahman, Sherihan Fathey, René Hurlemann, Mohamed E. G. Elsayed, Joshuan J. Barboza, Aroop Mohanty, Alfonso J. Rodriguez-Morales, Bijaya Kumar Padhi, Ranjit Sah

**Affiliations:** 1Faculty of Medicine, Aleppo University, Aleppo 15310, Syria; 2Clinical Medicine, Weill Cornell Medical College, Al-Rayyan 36623, Qatar; 3NMC Royal Hospital, 16th Street, Khalifa, Abu Dhabi 35233, United Arab Emirates; Medical Research Division, Department of Internal Medicine, The National Research Centre, Cairo 11511, Egypt; 4Department of Internal Medicine, Syrian Private University, Damascus 20872, Syria; 5Faculty of Medicine of Algiers, University of Algiers 1, Alger Centre 16000, Algeria; 6Department of Neurosurgery, Qena University Hospitals, South Valley University, Qena 83511, Egypt; 7Department of Ophthalmology, University of California, Los Angeles, CA 90089, USA; 8Faculty of Medicine, Jordan University of Science and Technology, Ar-Ramtha 72764, Jordan; 9JLNM Hospital, Rainawari, Srinagar 190001, India; 10Directorate of Health Services, Jammu and Kashmir 184121, India; 11Department of Physical Therapy and Rehabilitation, Fenerbahçe Üniversitesi, Istanbul 34758, Turkey; 12University of Jordan, Amman 11110, Jordan; 13Faculty of Medicine, Sana′a University, Sana′a 1247, Yemen; 14MSH Statistics, Cairo 45785, Egypt; 15Internal Medicine Department, Faculty of Medicine, Zagazig University, Zagazig 44519, Egypt; 16Department of Health, Giza 12511, Egypt; 17Department of Psychiatry, School of Medicine and Health Sciences, Carl von Ossietzky University Oldenburg, 26122 Oldenburg, Germany; 18Department of Psychiatry, University Hospital Bonn, 53229 Bonn, Germany; 19Research Center Neurosensory Science, Carl von Ossietzky University Oldenburg, 26122 Oldenburg, Germany; 20Department of Psychiatry and Psychotherapy III, University of Ulm, Leimgrubenweg, 89075 Ulm, Germany; 21Escuela de Medicina, Universidad César Vallejo, Trujillo 13007, Peru; 22Department of Clinical Microbiology, All India Institute of Medical Sciences, Gorakhpur 273008, India; 23Grupo de Investigación Biomedicina, Faculty of Medicine, Fundación Universitaria Autónoma de las Américas—Institución Universitaria Visión de las Américas, Pereira 660003, Risaralda, Colombia; 24Clinical Epidemiology and Biostatistics, School of Medicine, Universidad Científica del Sur, Lima 4861, Peru; 25Gilbert and Rose-Marie Chagoury School of Medicine, Lebanese American University, Beirut P.O. Box 36, Lebanon; 26Department of Community Medicine and School of Public Health, Postgraduate Institute of Medical Education and Research, Chandigarh 160012, India; 27Department of Clinical Microbiology, Institute of Medicine, Tribhuvan University Teaching Hospital, Kathmandu 44600, Nepal; 28Department of Clinical Microbiology, DY Patil Medical College, Hospital and Research Centre, DY Patil Vidyapeeth, Pune 411000, India

**Keywords:** monkeypox, COVID-19, knowledge, vaccine adoption, fear, anxiety

## Abstract

Background: The outbreak of monkeypox was declared a global public health emergency by the World Health Organization on 23 July 2022. There have been 60,000 cases reported worldwide, most of which are in places where monkeypox has never been seen due to the travel of people who have the virus. This research aims to evaluate the general Arabic population in regard to the monkeypox disease, fears, and vaccine adoption after the WHO proclaimed a monkeypox epidemic and to compare these attitudes to those of the COVID-19 pandemic. Methods: This cross-sectional study was performed in some Arabic countries (Syria, Egypt, Qatar, Yemen, Jordan, Sudan, Algeria, and Iraq) between 18 August and 7 September 2022. The inclusion criteria were the general public residing in Arabic nations and being older than 18. This questionnaire has 32 questions separated into three sections: sociodemographic variables, prior COVID-19 exposure, and COVID-19 vaccination history. The second portion assesses the knowledge and anxieties about monkeypox, while the third section includes the generalized anxiety disorder (GAD7) scale. Logistic regression analyses were performed to compute the adjusted odds ratios (aOR) and their confidence intervals (95%CI) using STATA (version 17.0). Results: A total of 3665 respondents from 17 Arabic countries were involved in this study. Almost two-thirds (*n* = 2427, 66.2%) of the participants expressed more worry about COVID-19 than monkeypox diseases. Regarding the major cause for concern about monkeypox, 39.5% of participants attributed their anxiety to the fear that they or a member of their family may contract the illness, while 38.4% were concerned about monkeypox becoming another worldwide pandemic. According to the GAD 7 score, 71.7% of the respondents showed very low anxiety toward monkeypox and 43.8% of the participants scored poor levels of knowledge about monkeypox disease. Participants with previous COVID-19 infection showed a 1.206 times greater acceptance to receive the monkeypox vaccine than those with no previous infection. A 3.097 times higher concern for monkeypox than COVID-19 was shown by the participants who perceived monkeypox as dangerous and virulent than those who did not. Participants who have a chronic disease (aOR: 1.32; 95%CI: 1.09–1.60); participants worried about monkeypox (aOR: 1.21; 95%CI: 1.04–1.40), and perceived monkeypox as a dangerous and virulent disease (aOR: 2.25; 95%CI: 1.92–2.65); and excellent knowledge level (aOR: 2.28; 95%CI: 1.79–2.90) have emerged as significant predictors. Conclusions: Our study reported that three-fourths of the participants were more concerned about COVID-19 than monkeypox disease. In addition, most of the participants have inadequate levels of knowledge regarding monkeypox disease. Hence, immediate action should be taken to address this problem. Consequently, learning about monkeypox and spreading information about its prevention is crucial.

## 1. Introduction

The outbreak of monkeypox was declared a global public health emergency condition by the World Health Organization (WHO) on 23 July 2022. The majority of the reported cases were from Europe [1]. Since the beginning of the COVID-19 epidemic, nearly two-million fatalities have occurred worldwide. Extensive research has shown a correlation between COVID-19 and mental health problems, such as anxiety, depression, and mental distress. This association increased as the worldwide number of COVID-19 cases and fatalities increased [2]. The monkeypox virus is a double-stranded DNA virus that can spread from animals to humans. It is in the genus Orthopoxvirus and the family Poxviridae [3]. There are approximately 21,504 cases in the US, but only 35 cases have been reported from Arabic countries [4]. Multiple animal hosts for this virus have been discovered, including rope squirrels, tree squirrels, Gambian pouched rats, dormice, and many kinds of monkeys [5]. Transmission is considered to occur by exposure to infected animals or people (animal to human/human to human). Body fluids, respiratory excretions, contact with cutaneous or mucosal lesions, and exposure to contaminated items are all transmission tracts for this disease [1,2]. The predicted incubation time for monkeypox ranges between 5 and 21 days. The initial illness signs are vague, including fever, chills, headaches, lymphadenopathy, back pain, myalgia, and, eventually, the appearance of a rash [6]. This disease has been linked to several complications, including bacterial superinfection of the skin, pneumonia, encephalitis, sepsis, and death [7]. Psychiatric and neurological manifestations might also occur [8]. In most cases, monkeypox infections heal without medical treatment. Oral or intravenous rehydration is advised to maintain hydration levels in patients with digestive symptoms, such as vomiting and diarrhea. Several antiviral drugs, including tecovirimat, brincidofovir, and cidofovir, have been shown to be effective against the monkeypox virus [9]. There has been evidence of the vaccine′s effectiveness in protecting against the monkeypox virus. Modified vaccinia Ankara and ACAM2000 are the two vaccines that have been established [10]. Recent research has shown that the general public is more concerned about COVID-19 than about monkeypox. Deaths from monkeypox have not been documented in areas where the disease is not naturally present, lending credence to the idea that the current outbreak′s clinical repercussions are less severe than those seen in endemic areas [11,12]. However, there is still a gap in the literature of the assessment of the knowledge and the psychological behaviors toward the monkeypox illness and whether there is a more significant worry about it as the number of recorded cases increases. This research aims to evaluate Arabic people′s perspectives on monkeypox disease, their fears, and their vaccine adoption after the WHO proclaimed a monkeypox epidemic, and to compare these attitudes to those of the COVID-19 pandemic.

## 2. Methods

### 2.1. Study Design

A cross-sectional online study was performed in 17 Arabic countries (Syria, Egypt, Qatar, Yemen, Jordan, Sudan, Algeria, Iraq, and other countries) between 18 August and 7 September 2022, to examine the Arabic people′s perspectives on the monkeypox disease, their fears, and their vaccine adoption, and to compare these attitudes to those of the COVID-19 pandemic. The participants were required to be individuals of the general public residing in Arabic nations and older than 18 years. Medical professionals, students, and staff were not eligible for participation. All of the participants were informed of the study′s goals, the researchers′ identities, their ability to opt out of participation, the confidentiality and security of their data, and the importance of providing all of the requested information. This questionnaire was constructed using a complete, verified scale based on previous research [11,13]. The survey was then translated from English into Arabic for the respondent′s comprehension by a native translator. Due to concerns about security, a Google form survey was developed and distributed over social media sites, such as Facebook, WhatsApp, and Telegram. In addition, in the individual governorates, retail malls, parks, public squares, and other public meeting areas were accessible for data collection, as were face-to-face interviews.

### 2.2. Sample Size Calculation

Depending on the Arab population count (https://data.worldbank.org/indicator/SP.POP.TOTL?locations=1A, accessed on 1 December 2021), and using Calculator.net (https://www.calculator.net/sample-size-calculator.html, accessed on 1 December 2021), we conducted a statistical power analysis for sample size calculation using a population proportion of 50%, a margin of error of 0.05, and a degree of confidence of 99%, so the minimum sample size was 385. The overall sample size was 3822 participants.

### 2.3. Measures

This questionnaire has 32 questions separated into three sections: sociodemographic factors, prior COVID-19 exposure, and COVID-19 vaccination history. The second portion assesses the participants’ knowledge and anxieties about Monkeypox, while the third section includes the Generalized Anxiety Disorder (GAD7) scale in reference to the monkeypox virus [14]. The utilized questionnaire is supplied as a “Appendix A”.

### 2.4. Sociodemographic Characteristics

This section contains six questions regarding the participant′s sociodemographic characteristics, including age, gender, place of residence, marital status, economic status, and chronic illness status (hypertensive, diabetes, etc.). This section also includes three questions about the participants′ prior exposure to COVID-19, including whether or not they have ever been infected with the virus, whether or not they took preventative measures against contracting the disease during the COVID-19 pandemic, and whether or not they are more concerned about a possible outbreak of monkeypox than the COVID-19 epidemic.

### 2.5. Knowledge and Concerns about Monkeypox

There are 11 multiple-choice questions here on the monkeypox virus that participants may answer yes, no, or not sure to gauge their level of familiarity with Monkeypox. The participants were asked whether they believed that monkeypox was caused by a virus or bacteria, whether it could be spread from person to person, whether its symptoms were similar to those of smallpox, whether its rash and papules were indicative of the disease, and whether antibiotics were effective against the virus. Extensive prior research uses of this instrument attest to its reliability [13]. There are also three questions in this section that assess how worried people are about monkeypox and related issues (such as whether or not the virus is serious enough to warrant taking precautions to avoid spreading it and whether or not people should be quarantined worldwide). In addition, this section includes two questions to assess the participants’ willingness to obtain the monkeypox vaccination and to identify the most probable receivers. This question was extracted from a previously verified survey [11].

### 2.6. Generalized Anxiety Disorder (GAD) toward Monkeypox

This tool consists of seven validated questions to assess the GAD of the participants towards the monkeypox virus [14]. In this tool, the respondents were asked to assess the frequency with which they had experienced symptoms such as worry, concern, restlessness, impatience, and dread during the previous two weeks. We gave values between 0 and 3 to the following frequency categories: never, sometimes, often, and very frequently. The GAD7 scores were tallied and classified as minimal (0–4), mild (5–9), moderate (10–14), and severe (0–14). (15–21).

### 2.7. Pilot Study

We administered the questionnaire to 50 Arabic individuals, selected randomly to ensure its validity and readability, and we detected high levels of internal consistency (Cronbach′s alpha varied from 0.712 to 0.861). In addition, we modified the questionnaire according to the individuals′ feedback. However, these 50 responses were excluded from the sample that was eligible for the analysis.

Ethical Consideration: Ethical Approval is taken from the Syrian Arab Republic, Aleppo city, Faculty of Medicine, Aleppo University. The IRB approval number is SA-AL 4853B.

The Syrian Ethical Society for Scientific Research provided its stamp of approval (SA-AL4853B). Furthermore, at least one ethical approval was taken from each of the inquired country in our study. The participants were given a Google Survey URL and asked to confirm their agreement to participate in the study on the first page of the survey. There is a wealth of background reading for the participants to peruse before they fill out the questionnaire on the next page and complete this survey within 5–12 min. The responses were collected and kept in an encrypted internet database.

### 2.8. Statistical Analysis

The statistical data analysis was conducted using the IBM SPSS statistical analysis tool (28). Means and standard deviations were used in the process of describing the continuous variables, while frequencies and percentages were utilized in the process of describing the categorically measured variables. Multivariate Binary Logistic Regression Analysis was used so that the investigators could determine which variables could be responsible for people′s concerns over monkeypox and their willingness to be vaccinated against the disease. An odds ratio (OR) with accompanying 95% confidence intervals was used to represent the connection between the predictors and the outcome-dependent variables in the multivariate Logistic Binary regression analysis. *p*-value ≤ 0.05 was considered a statistically significant value.

## 3. Results

The demographic variables of the respondents are summarized in Table 1. A total of 3772 responses from 17 Arabic countries were collected in this study; however, the final eligible sample size for data analysis, after omitting the missing data, was 3665. The highest proportion of the inquired participants was from Yemen (*n* = 689, 18.8%), followed by Syria (*n* = 394, 10.8%). More than half of the participants were female (2187, 59.7%). Most of the respondents (*n* = 3016, 82.3%) were aged between 18–34. Nearly two-thirds of the study participants were single (2249, 61.4%). Regarding their economic status, 56.1% (*n* = 2055) of the respondents had a moderate economic condition, while 30.6% (*n* = 1123) had a good economic condition. The majority of the participants (84.0%, *n* = 3078) had no history of chronic disease.

Regarding Table 2, 45.8% (*n* = 1679) of the participants reported a previous infection of the COVID-19 disease; additionally, about half of the participants (51.9%, *n* = 1903) reported a medium commitment during the COVID-19 pandemic. Almost two-thirds (*n* = 2427, 66.2%) of the participants expressed more worry about COVID-19 than monkeypox diseases. Regarding the major cause for concern about Monkeypox, 39.5% of the participants attributed their anxiety to the fear that they or a family member may contract the illness, while 38.4% were concerned about another worldwide pandemic being caused by Monkeypox. Overall, 69.9% of the respondents agreed that Monkeypox is a dangerous and virulent illness that requires respiratory and contact precautions. According to the GAD 7 score, 71.7% of the respondents showed very low anxiety toward monkeypox. However, 43.8% of the study sample scored poor levels of knowledge regarding monkeypox disease (Table 2).

Ten out of fourteen of the predicted variables were statistically significantly associated with more willingness to vaccinate against Monkeypox (*p* < 0.05). The respondents aged between 35–44 years had fewer odds of willingness to receive the monkeypox vaccine compared to the 18–34 age group (OR = 0.715). In comparison to those who were single, married respondents were projected to have less acceptance of the monkeypox vaccination (OR = 0.855). The participants with excellent income were 2.069 times more likely to receive the vaccine than those with bad income. Participants who had a previous COVID-19 infection showed a 1.206 times greater acceptance to receive the monkeypox vaccine than those with no previous infection. Regarding the main reasons for worry about Monkeypox disease, the participants who expressed worry that Monkeypox might surge to cause a national lockdown, who were concerned that an international flight suspension would happen, and who reported other reasons of worry had a significant drop in vaccination acceptability, by 0.016, 0.027, and <0.001 times, respectively, than those who expressed concern that they or a member of their family could contract the illness. A 1.889 times higher agreement for the monkeypox vaccine was expected by the participant who perceived Monkeypox as dangerous and virulent more than those who did not (Table 3).

Eight out of fourteen of the predictor factors were significantly linked with greater worry about Monkeypox than COVID-19 (*p*-value < 0.05). Females were more 1.379 times more likely to be concerned about Monkeypox than males. Participants aged 35–44 had low odds of worry about Monkeypox compared to those 18–34. Low worries were predicted among the participants with chronic disease, 0.731 times more than those without. Regarding the main reasons for worry about the Monkeypox disease, the participants who expressed worry that Monkeypox might surge to cause another worldwide pandemic had a significant drop, by 0.723 times, in worry about Monkeypox than those who expressed concern that they or a member of their family could contract the illness. A 3.097 times higher concern for Monkeypox than COVID-19 was shown by the participants who perceived Monkeypox as dangerous and virulent than those who did not (Table 4).

Table 5 depicts the multiple logistic regression analysis findings for the association between the intention to vaccinate and the socio-behavioral characteristics of the study participants. The participants who have a chronic disease (aOR: 1.32; 95%CI: 1.09–1.60), participants worried about Monkeypox (aOR: 1.21; 95%CI: 1.04–1.40), those who perceived monkeypox as a dangerous and virulent disease (aOR: 2.25; 95%CI: 1.92–2.65), and those with an excellent knowledge level (aOR: 2.28; 95%CI: 1.79–2.90) emerged as significant predictors.

## 4. Discussion

The recent spread of global health pandemics is one of the major problems facing humanity and affecting the world in various aspects of life [15]. The COVID-19 pandemic during the last three years is the best example of the extent to which humanity might be affected by such outbreaks. Accordingly, the recent announcement by the WHO of the possibility of facing the threat of a new global pandemic (i.e., Monkeypox (MPX) as a Public Health Emergency of International Concern) [16] during our confrontation with the COVID-19 pandemic encourages the world to be vigilant and take the necessary precautionary measures to avoid the recurrence of the tragedy of the Corona-virus pandemic and other previous intractable pandemics. However, this caution may result in episodes of anxiety and depression of varying degrees among the mass of people that may repeat what they have experienced during the harsh conditions of the COVID-19 pandemic [17,18,19,20].

In our cross-sectional online survey, we assessed the level of anxiety among a sample of the Arab population following the Monkeypox virus (MPXV) prevalence in some Arab countries. Less than half (45.8%) of the participants confirmed a COVID-19 infection history, while slightly more than half (51.9%) reported a medium commitment and average rules followed during the COVID-19 pandemic. This constant compliance with coronavirus control measures is crucial for the continued education and awareness messages concerning the Monkeypox disease. Despite the increase in MPX cases in Arab countries [4], very low anxiety concerning this rising health problem was recorded among most participants (71.7%) according to the GAD7 score. This may be attributed to people′s low level of knowledge about the disease, as we found that 43.8% of the participants recorded a poor level of knowledge about the disease.

The small number of cases of Monkeypox and the rarity of men who engage in male-to-male sex in Arabic-speaking nations may have impacted our findings by lowering people′s awareness and fear of the disease. Accordingly, it is necessary to conduct awareness campaigns about its transmission methods, as well as prevention and management for the public through the media and in schools at different stages. In addition, with the world still facing the COVID-19 pandemic at present and since 2020, we compared the levels of public worry about the MPX epidemic with that of the COVID-19 pandemic.

We found that most participants (66.2%) worried more about COVID-19 than MPX. However, more concern and worry about MPX was paid among some groups of the participants than their equivalents (e.g., females (OR = 1.379), participants aged 18–34, participants without a chronic disease (OR = 1.368), more details in the results). The COVID-19 mental disorders collaborators reported similar findings in their large systematic review about the prevalence of major depressive disorders and anxiety disorders induced by the COVID-19 pandemic [17]. However, the reasons for anxiety about MPX differ from those for COVID-19. We found that most of the anxiety in our study stemmed from their fear that they or a member of their family might be infected with MPXV or worry about another worldwide pandemic and its consequences. In contrast, the study population reported a fear of international flight suspension as the less worrying factor. The major worries during COVID-19 were the high mortality rates and morbidities associated with the disease [17,21].

Although no specific vaccine protects against MPXV, many orthopoxvirus vaccines have been found to cross-protect against MPXV by antigenic similarity [22,23]. Smallpox vaccines have great efficacy in preventing MPX infection, with a clinical effectiveness of about 85% [24]. According to the CDC, two smallpox vaccines (ACAM2000 AND JYNNEOS) are approved to protect against both smallpox and MPX in the USA [25]. Monkeypox vaccination can be either pre-exposure or post-exposure immunization [22,24,25]. While a pre-exposure vaccination can reduce the risk of catching the infection, a post-exposure vaccination within 4–14 days can ameliorate the symptoms of the disease and help with rapid recovery [24].

Based on our analysis, a large percentage of the respondents to our survey showed acceptance of the MPXV vaccination. Thus, an effective measure that can help control the outbreak in the Arab world is the implementation of vaccination for high-risk groups and the ring vaccination (i.e., vaccination of direct contacts of confirmed infected people [24,26]. Taking this into consideration, the previous experience of the COVID-19 pandemic stresses the importance of implementing strong vaccination strategies. Based on Watson et al.’s analysis of the global impact induced by COVID-19 vaccination, millions of deaths were avoided after the vaccination, with an estimated global reduction of 63% of the total deaths within the first year of vaccination [27]. With a major global event such as the 2022 FIFA World Cup in Qatar, strict precautions must be taken to avoid the spread of MPX and thus to avoid facing the risk of a new global pandemic, in addition to the current COVID-19 one [28].

This might include: (1) raising awareness and knowledge about the disease through social media and educational campaigns; (2) conducting screening tests for travelers before entering the country; (3) emphasizing the ability of the healthcare system to manage the situation; (4) the rapid diagnosis and control of any new cases; and (5) taking the MPXV vaccination strategy into consideration [29].

Finally, Ennab et al. and Sah et al. presented some of the commandments and lessons learned from the COVID-19 pandemic, which we can benefit from to reduce the risk of facing a new health crisis [30,31].

The concern and worry of the Arab population in the MENA region being average, reasonable, and not exaggerated should not lead to neglect that will result in a new pandemic. Whilst the Monkeypox infection appears to cause less mortality than the COVID-19 infection, the symptoms tend to be more severe and difficult. Moreover, the world health authorities are already weakened by the coronavirus pandemic, which imposes a strong preparation for a probable second pandemic in the 21st century caused by Monkeypox [21].

More global future studies are needed to determine the specific concerns about the Monkeypox virus pandemic and to define the knowledge gaps towards the Monkeypox virus, which helps the community to properly face any predicted pandemic. Strengths and limitations.

Although cross-sectional studies are inexpensive and can be easily accomplished, and analytical cross-sectional surveys may be used to study the relationship between a potential risk factor and a health result, the credibility of the assumptions that can be made about the relationship between a risk factor and a health result from this kind of research is limited. Furthermore, those who live in distant areas, do not have access to the internet, or are elderly will not be able to participate in the research by completing the online survey. In our study, an investigator was assigned to each country that we asked about to continuously review the data collection procedure and omit the random and multiple auto-responses. In addition, we did not include healthcare professionals or medical students. Delivering the questionnaire to people with no internet services, people with no smartphones, and people in inadequate internet and electricity service areas were important challenges during this study.

## 5. Conclusions

Overall, our results have shown knowledge gaps of the Monkeypox virus among the general population in Arabic countries, especially among people who have previously been affected by COVID-19; however, those with good knowledge of Monkeypox show lower levels of concern about new cases of Monkeypox in Arabic countries than those for COVID-19. The two most significant determinants, the rarity of males who partake in male-to-male sex in Arabic-speaking regions and the low number of monkeypox cases, may have affected our results by reducing people′s knowledge and fear of the virus. Healthcare providers should be knowledgeable about raising the knowledge of Monkeypox among the general population, specifically the signs and symptoms and the available diagnostic methods for this disease. Permanent screening programs should be conducted for people coming from endemic areas and people with high-risk infection levels, such as the LGBTQ community.

In addition, the historical experience of the COVID-19 pandemic highlights how essential it is to adopt effective vaccination and preventive policies.

## Figures and Tables

**Table 1 vaccines-11-00759-t001:** Sociodemographic characteristics of participants (*n* = 3665).

Variable	Frequency	Percentage
Country		
Yemen	689	18.8%
Algeria	335	9.1%
Syria	394	10.8%
Jordan	801	21.9%
Egypt	577	15.7%
Sudan	444	12.1%
Qatar	43	1.2%
Saudi Arabia	107	2.9%
Iraq	127	3.5%
Oman	18	0.5%
United Arab Emirate	54	1.5%
Lebanon	7	0.2%
Somalia	19	0.5%
Palestine	12	0.3%
Kuwait	25	0.7%
Libya	7	0.2%
Morocco	6	0.2%
Sex		
Female	2187	59.7%
Male	1477	40.3%
Age Groups		
18–34 years	3016	82.3%
35–44 years	414	11.3%
45–54 years	141	3.8%
55—64 years	94	2.6%
Marital Status		
Single	2249	61.4%
Married	1321	36.0%
Divorced	53	1.4%
Widowed	42	1.1%
Residence		
Countryside	818	22.3%
City	2847	77.7%
Income		
Bad	191	5.2%
Moderate	2055	56.1%
Good	1123	30.6%
Excellent	296	8.1%
Chronic Disease		
No	3078	84.0%
Yes	587	16.0%

**Table 2 vaccines-11-00759-t002:** Respondents′ attitudes, perceptions, and beliefs about Monkeypox disease (*n* = 3665).

Variable	Frequency	Percentage
Have you been affected by the COVID-19 disease?		
No	1986	54.2%
Yes	1679	45.8%
Participant′s and family members compliance with COVID-19 pandemic precautions		
Rarely committed	602	16.4%
Medium commitment	1903	51.9%
Always committed	1160	31.7%
Worry from Monkeypox compared to COVID-19		
Much worried with the COVID-19	2427	66.2%
More worried with Monkeypox	1238	33.8%
Main reasons for your worry from Monkeypox disease		
Self and family worry of Monkeypox infection	1447	39.5%
Worry from another worldwide pandemic	1409	38.4%
Worried that Monkeypox might surge to cause a national lockdown	400	10.9%
Concerned that an international flight suspension happens	79	2.2%
Other worries	330	9.0%
Do you perceive Monkeypox as a dangerous and virulent disease that calls for respiratory and contact precautions?		
No	1102	30.1%
Yes	2563	69.9%
GAD7 Score, Mean (SD)	3.15 (4.46)	–
GAD7 Score Levels		
Very low Anxiety	2627	71.7%
Mild Anxiety	621	16.9%
Moderate Anxiety	321	8.8%
High Anxiety	96	2.6%
Overall Monkeypox Disease Knowledge score, mean (SD). Maximum Score = 11	3.99 (2.87)	-
Overall Monkeypox Disease Knowledge Score Levels		
Poor	1604	43.8%
Good	1668	45.5%
Excellent	393	10.7%

**Table 3 vaccines-11-00759-t003:** Multivariate Binary Logistic Regression Analysis of people′s odds of agreement to vaccinate against Monkeypox Disease, (Agree versus Disagree).

Predictor	Adjusted Odds Ratio	95% Confidence Interval		*p* Value
		Lower	Upper	
Sex				
Female–Male	0.857	0.731	1.005	0.058
Age Groups				
35–44–18–34 years	0.715	0.555	0.922	0.01 *
45–54–18–34 years	0.612	0.408	0.918	0.018 *
55–64–18–34 years	0.843	0.527	1.348	0.475
Marital Status				
Married–Single	0.855	0.718	1.019	0.08
Divorced–Single	0.6	0.331	1.091	0.094
Widowed–Single	0.952	0.481	1.885	0.887
Residence				
City–Countryside	0.81	0.678	0.968	0.021 *
Income				
Moderate–Bad	1.159	0.834	1.611	0.38
Good–Bad	1.278	0.905	1.805	0.163
Excellent–Bad	2.069	1.371	3.122	<0.001 *
Chronic Disease				
Yes–No	1.193	0.976	1.458	0.085
Previous COVID-19 infection				
Yes–No	1.206	1.031	1.411	0.019 *
Participant and family members compliance				
Medium commitment–Rarely committed	0.871	0.711	1.066	0.181
Always committed–Rarely committed	1.012	0.804	1.275	0.916
Worry from Monkeypox compared to COVID19				
More worried with Monkeypox–Much worried with the COVID-19	1.111	0.948	1.302	0.193
Main reasons for your worry from Monkeypox disease				
Worry from another worldwide pandemic–Self and family worry of Monkeypox infection	0.949	0.809	1.115	0.526
Worried that Monkeypox might surge to cause a national lockdown–Self and family worry of Monkeypox infection	0.74	0.579	0.945	0.016 *
Concerned that an international flight suspension happens–Self and family worry of Monkeypox infection	0.56	0.335	0.935	0.027 *
Other worries–Self and family worry of Monkeypox infection	0.461	0.349	0.607	<0.001 *
Do you perceive Monkeypox as a dangerous and virulent				
Yes–No	1.889	1.591	2.243	<0.001 *
Total Knowledge Score	1.099	1.07	1.128	<0.001 *
Total GAD Score	1.045	1.028	1.064	<0.001 *

* *p* value significant if it is equal to or less than 0.05.

**Table 4 vaccines-11-00759-t004:** Multivariate Binary Logistic Regression Analysis of respondents′ odds of higher worry level from Monkeypox disease compared to COVID-19.

Predictor	Adjusted Odds Ratio	95% Confidence Interval		*p* Value
		Lower	Upper	
Sex				
Female–Male	1.379	1.1611	1.638	<0.001 *
Age Groups				
35–44–18–34 years	0.721	0.5453	0.953	0.022 *
45–54–18–34 years	0.964	0.6228	1.491	0.867
55–64–18–34 years	0.977	0.573	1.666	0.932
Marital Status				
Married–Single	1.177	0.9779	1.418	0.085
Divorced–Single	0.814	0.4218	1.571	0.54
Widowed–Single	1.417	0.6728	2.984	0.359
Residence				
City–Countryside	0.921	0.761	1.115	0.398
Income				
Moderate–Bad	0.834	0.5879	1.182	0.307
Good–Bad	0.857	0.5936	1.237	0.409
Excellent–Bad	1.058	0.6877	1.627	0.798
Chronic Disease				
Yes–No	0.731	0.5881	0.909	0.005 *
Previous COVID-19 infection				
Yes–No	0.898	0.759	1.062	0.21
Participant and family members compliance				
Medium commitment–Rarely committed	1.205	0.9622	1.508	0.104
Always committed–Rarely committed	0.968	0.7523	1.245	0.798
Main reasons for your worry from Monkeypox disease				
Worry from another worldwide pandemic–Self and family worry of Monkeypox infection	0.723	0.6091	0.858	<0.001 *
Worried that Monkeypox might surge to cause a national lockdown–Self and family worry of Monkeypox infection	1.07	0.8248	1.388	0.61
Concerned that an international flight suspension happens–Self and family worry of Monkeypox infection	0.869	0.5023	1.502	0.614
Other worries–Self and family worry of Monkeypox infection	1.038	0.775	1.39	0.802
Do you perceive Monkeypox as a dangerous and virulent				
Yes–No	3.097	2.5221	3.803	<0.001 *
Do you agree to be vaccinated by monkeypox vaccine				
No–Yes	1.107	0.9437	1.298	0.212
Total Knowledge Score	1.066	1.0367	1.096	<0.001 *
Total GAD Score	1.114	1.0953	1.134	<0.001 *

* *p* value significant if it is equal to or less than 0.05.

**Table 5 vaccines-11-00759-t005:** Multiple logistic regression analysis for the association between intention to vaccinate and socio-behavioral characteristics of study participants (*n* = 3664).

Variable	aOR [95% CI]	*p*-Value
Age (Reference: 18–34)		
35–44	0.65 [0.51, 0.83]	0.00 *
45–54	0.59 [0.40, 0.87]	0.01 *
55—64	0.86 [0.54, 1.36]	0.51
Gender (Reference: male)		
Female	1.08 [0.93, 1.24]	0.31
Marital Status (Reference: Single)		
Married	0.88 [0.74, 1.03]	0.11
Divorced	0.76 [0.42, 1.36]	0.35
Widowed	0.99 [0.51, 1.93]	0.97
Residence (Reference: Countryside)		
City	0.76 [0.65, 0.90]	0.00 *
Chronic Disease (Reference: No)		
Yes	1.32 [1.09, 1.60]	0.00 *
Previous COVID19 Infection (Reference: No)		
Yes	1.08 [0.94, 1.23]	0.31
Participant′s and family members compliance with COVID-19 pandemic precautions (Reference: Rarely committed)		
Medium commitment	0.83 [0.69, 1.01]	0.06
Always committed	0.88 [0.71, 1.08]	0.22
Worry from Monkeypox compared to COVID-19 (Reference: Much worried with the COVID-19)		
More worried with Monkeypox	1.21 [1.04, 1.40]	0.01*
Do you perceive Monkeypox as a dangerous and virulent disease that calls for res (Reference: No)		
Yes	2.25 [1.92, 2.65]	0.00 *
Knowledge of Monkeypox (Reference: Poor)		
Good	1.62 [1.39, 1.87]	0.00 *
Excellent	2.28 [1.79, 2.90]	0.00 *

* *p* value significant if it is equal to or less than 0.05.

## Data Availability

The analyzed data is available for the readers upon responsible request from the first author or the corresponding author.

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
