# Peer review of "Monkeypox Post-COVID-19: Knowledge, Worrying, and Vaccine Adoption in the Arabic General Population"

_vaccines, 2023, doi:10.3390/vaccines11040759_

Round 1

Reviewer 1 Report

1. Abstract. The objective described in line 58-59 is repeated in Methods

2. Introduction. Please explain the relationship from ref 11 and 13. eg if they provided preliminary data from an earlier study to the current research.

3. Method. 

paragraph 2.1. Please confirm if medical healthcare was involved in communicating the questionaire. Especially the section on GAD. How was security of information protected using Google?

paragraph 2.7 What is the relevance of the 50 individuals? Their results should be included in the text.

Paragraph 2.8 Statistical analysis. What is considered as significant with the p=value?

4. Results. The definition of the variable  should be better defined. eg in Table 2, what is the definition of rarely committed.? Is the definition part of the questionaire? If appropriate, would propose adding a copy of the questionaire as a supplementary materials. The significance of p-value should also be described in the tables or method.

Table 5. is the result for p value of 0.00 acceptable? What do we interpret from here?

5. Discussion

Generally too long and should be paragraphed for easier reading.

Some points can be placed in the conclusion, which seemed kind of short.

Should address challenges encountered in the study.  Or limitations faced by investigators.

If the elderly populations has no access to internet, how can they be contacted? Is it important to contact this population since the world generally has an aging population?

Is monkeypox a relevant disease to be compared to COVID 19? How about influenza as it employs the same transmission route and may be a better virus to compare instead of monkeypox?

How would future similar study be addressed?

What differernce does this research made to the healthcare personnel or adminstrators in handling future pandemics? Is there something that the investigators would like to change? eg policies. This can be cited in the manuscript.

Author Response

We have attached the word file for a detailed response for the comments from the reviwers. 

Reviewer 2 Report

This manuscript surveyed the knowledge, worries, and vaccine adoption about monkeypox in the Arabic general population. Almost two third of participants were more worried about COVID-19 than monkeypox diseases, and the knowledge of monkeypox was generally low. Participants with previous COVID-19 infection showed greater acceptance to receive the monkeypox vaccine. The results of the survey provide insight into the concerns about the monkeypox virus and its prevention.

 Here are some suggestions that may improve this work.

1. More detailed information about the spread of the monkeypox virus should be provided, as the virus can be transmitted through contact and air. The misunderstanding about virus transmission needs to be clarified. 

2. The authors need to emphasize the similarities and differences between the symptoms of the monkeypox virus and other common infectious diseases and remind people to detect the monkeypox virus when the symptoms are suspected.  

3. The manuscript contains a considerable number of unclarities and errors that need correction to increase readability.

Author Response

WE have attached a word file containing the detailed response to the comments received from the reviewers. 
